# Decoding and reprogramming fungal iterative nonribosomal peptide synthetases

Dayu Yu[1,2,*], Fuchao Xu[2,*], Shuwei Zhang[2] & Jixun Zhan[2]

Nonribosomal peptide synthetases (NRPSs) assemble a large group of structurally and functionally diverse natural products. While the iterative catalytic mechanism of bacterial NRPSs is known, it remains unclear how fungal NRPSs create products of desired length. Here we show that fungal iterative NRPSs adopt an alternate incorporation strategy. Beauvericin and bassianolide synthetases have the same $C_1-A_1-T_1-C_2-A_2-MT-T_{2a}-T_{2b}-C_3$ domain organization. During catalysis, $C_3$ and $C_2$ take turns to incorporate the two biosynthetic precursors into the growing depsipeptide chain that swings between $T_1$ and $T_{2a}/T_{2b}$ with $C_3$ cyclizing the chain when it reaches the full length. We reconstruct the total biosynthesis of beauvericin *in vitro* by reacting $C_2$ and $C_3$ with two SNAC-linked precursors and present a domain swapping approach to reprogramming these enzymes for peptides with altered lengths. These findings highlight the difference between bacterial and fungal NRPS mechanisms and provide a framework for the enzymatic synthesis of non-natural nonribosomal peptides.

[1] Department of Applied Chemistry and Biological Engineering, College of Chemical Engineering, Northeast Electric Power University, Jilin, Jilin 132012, China. [2] Department of Biological Engineering, Utah State University, 4105 Old Main Hill, Logan, Utah 84322, USA. * These authors contributed equally to this work. Correspondence and requests for materials should be addressed to J.Z. (email: jixun.zhan@usu.edu).

Nonribosomal peptides (NRPs) represent an important family of bioactive natural products, such as daptomycin, penicillin and vancomycin. They are assembled by nonribosomal peptide synthetases (NRPSs) that consist of a series of catalytic domains[1]. Adenylation (A), thiolation (T) and condensation (C) are essential domains for the formation and elongation of the peptide chain. Other tailoring domains, such as methyltransferase (MT), oxidation (Ox) and epimerization (E), contribute to the large structural diversity in natural NRPs[2]. Iterative NRPSs generate NRP products with repeated moieties/monomers[3], such as bacterial metabolites enterobactin, echinomycin and gramicidin, as well as fungal natural products beauvericin (**1**), beauvericins A–C (**2**–**4**), bassianolide (**5**), enniatins A–C (**6**–**8**) and verticilide (**9**) (Fig. 1a). These molecules possess a wide range of biological activities, including antimicrobial, insecticidal, anthelmintic, herbicidal, anti-haptotactic, anti-cholesterol and anticancer activities[3–8]. **1**–**9** are cyclic oligomers of monomer synthesized from a hydroxycarboxylic acid and an $N$-methylated amino acid. For instance, **1** is a cyclic trimer of a dipeptidol monomer synthesized from $N$-methyl-L-phenylalanine ($N$-Me-L-Phe) and D-hydroxyisovaleric acid (D-Hiv), and **5** is an octadepsipeptide containing four units of D-Hiv-$N$-Me-L-Leu (leucine: Leu). However, how fungal iterative NRPSs are programmed to assemble these compounds is not well understood.

*Beauveria bassiana* is a filamentous fungus known to produce **1**–**5**, and the enzymes responsible for the synthesis of these compounds have been reported[9,10]. The beauvericin and bassianolide synthetases (BbBEAS and BbBSLS) share significant sequence homology (66% identity and 79% similarity) and the same domain organization (Fig. 1b). In spite of the high sequence similarity, BbBEAS catalyses recursive head-to-tail condensation of three dipeptidol monomers, while BbBSLS condenses four monomers. Thus, these enzymes constitute a promising model to understand the product elongation and length control mechanism by fungal iterative NRPSs. The gene sequences of several other fungal NRPSs with the domain organization of $C_1$-$A_1$-$T_1$-$C_2$-$A_2$-MT-$T_{2a}$-$T_{2b}$-$C_3$ (Fig. 1b) have also been reported[11]. The common architecture of these NRPSs suggests that they share a general assembly rule.

Previous studies on bacterial iterative NRPSs, such as those involved in the biosynthesis of tyrocidine and gramicidin S, revealed that these enzymes use an oligomerization approach for chain elongation and length control through the C-terminal thioesterase (TE) domain. They oligomerize a monomer intermediate formed by upstream condensation domains and catalyse the subsequent head-to-tail cyclization for product release[12,13]. However, TE or a homologous domain is absent in fungal iterative NRPSs such as BbBEAS and BbBSLS. Based on the conserved signature regions, it was generally proposed that $A_1$ of BbBEAS and BbBSLS activates D-Hiv, while $A_2$ recognizes L-Phe and L-Leu, respectively. The activated D-Hiv is transferred to $T_1$, and MT methylates the $A_2$-activated amino acid[4]. However, the roles of several key domains in these fungal NRPSs remain unknown, including the twin $T_2$ domains and three C domains. More importantly, the product assembly process by fungal NRPSs is poorly understood.

In this work, we report that BbBEAS and BbBSLS adopt an alternate incorporation strategy to assemble the products. The two precursors, D-Hiv and $N$-Me-L-Phe/$N$-Me-L-Leu, are alternately incorporated into the extending depsipeptide chain by $C_2$ and $C_3$. The $C_3$ domain controls the product length and cyclizes the full-length depsipeptide chain to form the final products. Products with altered chain lengths are obtained by swapping the $C_3$ domain, providing a domain swapping approach to reprogram these enzymes for new molecules.

## Results

**Purification and reconstitution of BbBEAS and BbBSLS.** We have recently expressed BbBEAS and BbBSLS and achieved high-yield production of **1**–**5** ($33.8 \, \text{mg} \, \text{l}^{-1}$ for **1**–**4** and $21.7 \, \text{mg} \, \text{l}^{-1}$ for **5**) in *Saccharomyces cerevisiae* BJ5464-NpgA[14]. To decode the functions of the catalytic domains in BbBEAS (352 kDa) and BbBSLS (348 kDa), we first expressed and purified these giant enzymes from *S. cerevisiae* BJ5464-NpgA. As shown in Supplementary Fig. 1, both enzymes were expressed in the yeast and obtained in pure form ($0.8 \, \text{mg} \, \text{l}^{-1}$ for both BbBEAS and BbBSLS). Reaction of C-His$_6$-tagged BbBEAS with adenosine triphosphate (ATP), $S$-adenosylmethionine (SAM), L-Phe and D-Hiv yielded **1** (trace i, Fig. 1c), which was confirmed by electrospray ionisation mass spectrometry (ESI-MS) (Supplementary Fig. 2), high-resolution electrospray ionisation mass spectrometry (HR-ESI-MS) (Supplementary Fig. 3) and a comparison with the authentic sample prepared and fully characterized in our previous work (trace ii, Fig. 1c)[14]. Similarly, BbBSLS was found to synthesize **5** (traces iii and iv, Fig. 1c, Supplementary Fig. 2 and Supplementary Fig. 4) from L-Leu and D-Hiv[14]. Formation of **1** and **5** allowed direct functional characterization of BbBEAS and BbBSLS. These results indicated that both BbBEAS and BbBSLS were functionally expressed in *S. cerevisiae* and their catalytic activity can be reconstituted *in vitro*, providing a great platform to further study the catalytic domains in these modular NRPSs.

**Roles of the twin $T_2$ domains and the biosynthetic model.** Two possible biosynthetic models were proposed for BbBEAS and other similar NRPSs: linear (alternate incorporation of the precursors) and parallel (oligomerization of a monomer synthesized from the precursors) (Fig. 2a)[4]. In the linear model, $T_1$ and $T_{2a}$/$T_{2b}$ are alternately used for tethering the growing depsipeptide chain for addition of the D-hydroxycarboxylic acid and $N$-Me-L-amino acid units. $T_{2a}$ and $T_{2b}$ have the same function and only one of them is required. In contrast, in the parallel model, one of the twin $T_2$ domains is proposed to hold the synthesized dipeptidol monomer after it is synthesized at the other $T_2$ domain[4]. Additional monomers are added to the $T_{2b}$- or $T_{2a}$-linked monomer through oligomerization to form the dimer and subsequently the trimer (**1**) or tetramer (**5**). In this parallel model, both twin $T_2$ domains are required. Bacterial iterative NRPSs adopt a product assembly process that is more like the parallel model in Fig. 2a, except that the growing product chain is linked to the TE instead of a T domain. To understand what model is used by fungal iterative NRPSs, it is necessary to reveal the roles of the twin $T_2$ domains in BbBEAS and BbBSLS.

Sequence analysis revealed that the twin $T_2$ domains in both BbBEAS and BbBSLS contain the conserved motif (I/L)GG(D/H)SL, in which the key serine (Ser or S) residue serves as the phosphopantetheine attachment site (Supplementary Fig. 5). We first examined how inactivation of $T_{2a}$ and/or $T_{2b}$ affects the biosynthesis. Mutation of S2591 in $T_{2a}$ to alanine (A) did not abolish the production of beauvericins (trace i, Fig. 2b), but the titre ($0.7 \pm 0.1 \, \text{mg} \, \text{l}^{-1}$) calculated from three replicates was significantly lower than that ($33.8 \pm 1.4 \, \text{mg} \, \text{l}^{-1}$) by wild-type BbBEAS. In contrast, the $T_{2b}$ mutant BbBEAS-S2688A produced beauvericins (trace ii, Fig. 2b) with a titre of $23.0 \pm 1.0 \, \text{mg} \, \text{l}^{-1}$. When both key Ser residues were mutated, no products were synthesized by BbBEAS-S2591A/S2688A (trace iii, Fig. 2b).

We next tested how removal of the twin $T_2$ domains from BbBEAS affects beauvericin biosynthesis using a dissection and co-expression approach. BbBEAS was dissected at different positions and the fragments were functionally expressed in *S. cerevisiae* BJ5464-NpgA (traces i–iv, Fig. 2c). Using this system,

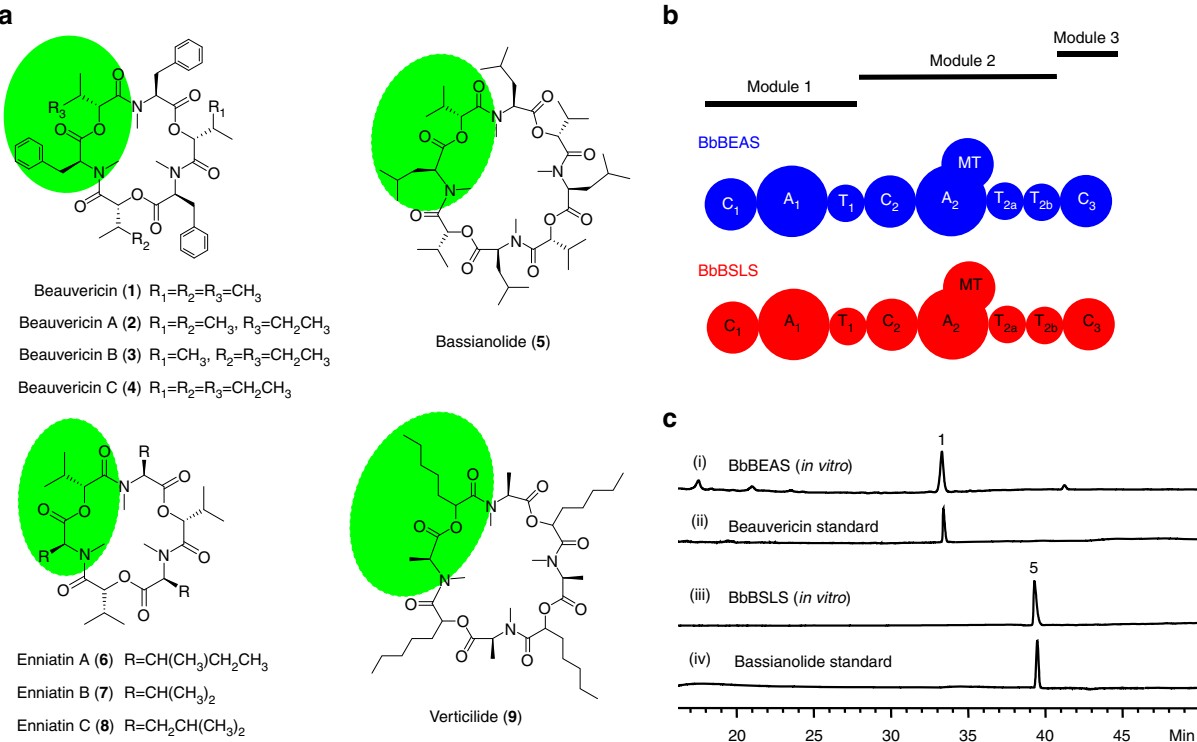

**Figure 1 | Fungal iterative NRPSs and their products. (a)** Representative NRPs (**1–9**) synthesized by fungal iterative NRPSs. The monomer unit is green shaded. (**b**) Domain organization of BbBEAS and BbBSLS. Both modular enzymes have the same architecture of $C_1$-$A_1$-$T_1$-$C_2$-$A_2$-MT-$T_{2a}$-$T_{2b}$-$C_3$. To differentiate these two enzymes, BbBEAS is shown in blue and BbBSLS in red. (**c**) HPLC analysis of the *in vitro* reaction of BbBEAS and BbBSLS with ATP, SAM, D-Hiv and L-amino acid (L-Phe for BbBEAS and L-Leu for BbBSLS).

we removed $T_{2a}$ from BbBEAS by co-expressing $C_1$-$A_1$-$T_1$-$C_2$-$A_2$-MT and $T_{2b}$-$C_3$ in the yeast. Liquid chromatography–mass spectrometry (LC–MS) analysis revealed that beauvericins were produced at $5.0 \pm 1.2$ mg l$^{-1}$ (trace v, Fig. 2c) in the absence of $T_{2a}$. This was validated by the construction of an intact enzyme BbBEAS-$\Delta T_{2a}$, which produced beauvericins at $0.5 \pm 0.1$ mg l$^{-1}$ (trace vi, Fig. 2c), confirming that the dissection and co-expression approach is effective and reliable. To remove $T_{2b}$, $C_1$-$A_1$-$T_1$-$C_2$-$A_2$-MT-$T_{2a}$ and standalone $C_3$ were co-expressed in the yeast, yielding **1–4** at $13.0 \pm 3.9$ mg l$^{-1}$ (trace vii, Fig. 2c). When both $T_2$ domains were removed, no products were detected (trace viii, Fig. 2c). These results were consistent with those from Ser mutations, which confirmed that only one of the twin $T_2$ domains is required for beauvericin biosynthesis.

Similarly, we also dissected BbBSLS before and after $T_{2a}$, and confirmed that the fragments can be co-expressed in the yeast to reconstitute bassianolide biosynthesis (traces i and ii, Fig. 2d). Removal of $T_{2a}$ (trace iii, Fig. 2d) or $T_{2b}$ (trace iv, Fig. 2d) did not abolish the production of **5**, although the titre by the former was much lower ($2.9$ mg l$^{-1}$ versus $10.6$ mg l$^{-1}$). This further confirmed that $T_{2a}$ plays a major role in the biosynthetic process as inactivation or removal of this domain significantly lowered the efficiency of the product assembly line. By contrast, $T_{2b}$ is less important in the biosynthetic process and may serve as an auxiliary T domain to contribute to the overall efficiency. Because the presence of both twin $T_2$ domains is not necessary for the biosynthesis, the possibility of oligomerization (parallel model) in chain elongation can be ruled out. Thus, it is concluded that **1–5** are assembled through an alternate incorporation approach (linear model, Fig. 2a) by the fungal NRPSs.

**The condensation activity of the $C_1$ and $C_2$ domains**. Another mystery of BbBEAS and BbBSLS is the unknown roles of the three

C domains in the biosynthetic process. We analysed the C domains in four reported fungal iterative NRPSs. As shown in Fig. 3a, the $C_2$ domains of these fungal NRPSs have the conserved HHxxxDG motif[15]. It was generally believed that the second histidine (H) in this motif serves as a general base for the condensation in most NRPSs, while aspartic acid (D) plays an important role in the structure of C domains[16,17]. The first H residue in this motif is not highly conserved, as some C domains such as the $C_3$ domains shown in Fig. 3a have an S at this position. The exact function of the G residue in this motif is not clear. Compared to $C_2$ and $C_3$, the $C_1$ domains of these fungal NRPSs are more divergent and $C_{1(\mathrm{BbBEAS})}$ does not have the essential H residue. As the C-terminal condensation domain, $C_3$ is likely involved in the final cyclization and concomitant release of the peptide chain from the end of the NRPS assembly lines. The remaining two condensation domains, $C_1$ and $C_2$, are possibly responsible for forming the ester and amide bonds in the product assembly line, respectively. We first tested the condensation activity of $C_2$. $C_{2(\mathrm{BbBEAS})}$ was overexpressed and purified from *E. coli* BL21(DE3) (Supplementary Fig. 6), and reacted with D-Hiv-SNAC (**S1**) (SNAC: *N*-acetylcysteamine)[18] and *N*-Me-L-Phe-SNAC (**S2**). As shown in Fig. 3b, D-Hiv-*N*-Me-L-Phe-SNAC (**S3**) was synthesized by $C_{2(\mathrm{BbBEAS})}$, and then spontaneously cyclized to form cyclo-D-Hiv-*N*-Me-L-Phe (**10**, Fig. 3b). In contrast, this domain could not form an ester bond as the reaction of **S1** and (D-Hiv-*N*-Me-L-Phe)$_2$-SNAC (**S4**) with $C_{2(\mathrm{BbBEAS})}$ did not yield any products (Supplementary Fig. 7). Consequently, the role of $C_2$ was determined to be forming the amide bond in the assembly process of NRPs.

We next attempted to understand whether $C_1$ forms the ester bond. $C_{1(\mathrm{BbBEAS})}$ was overexpressed and purified from *E. coli* BL21(DE3) (Supplementary Fig. 6). Incubation of $C_{1(\mathrm{BbBEAS})}$ with **S1** + **S2** or **S1** + **S4** did not yield any products (Fig. 3c,d). Because

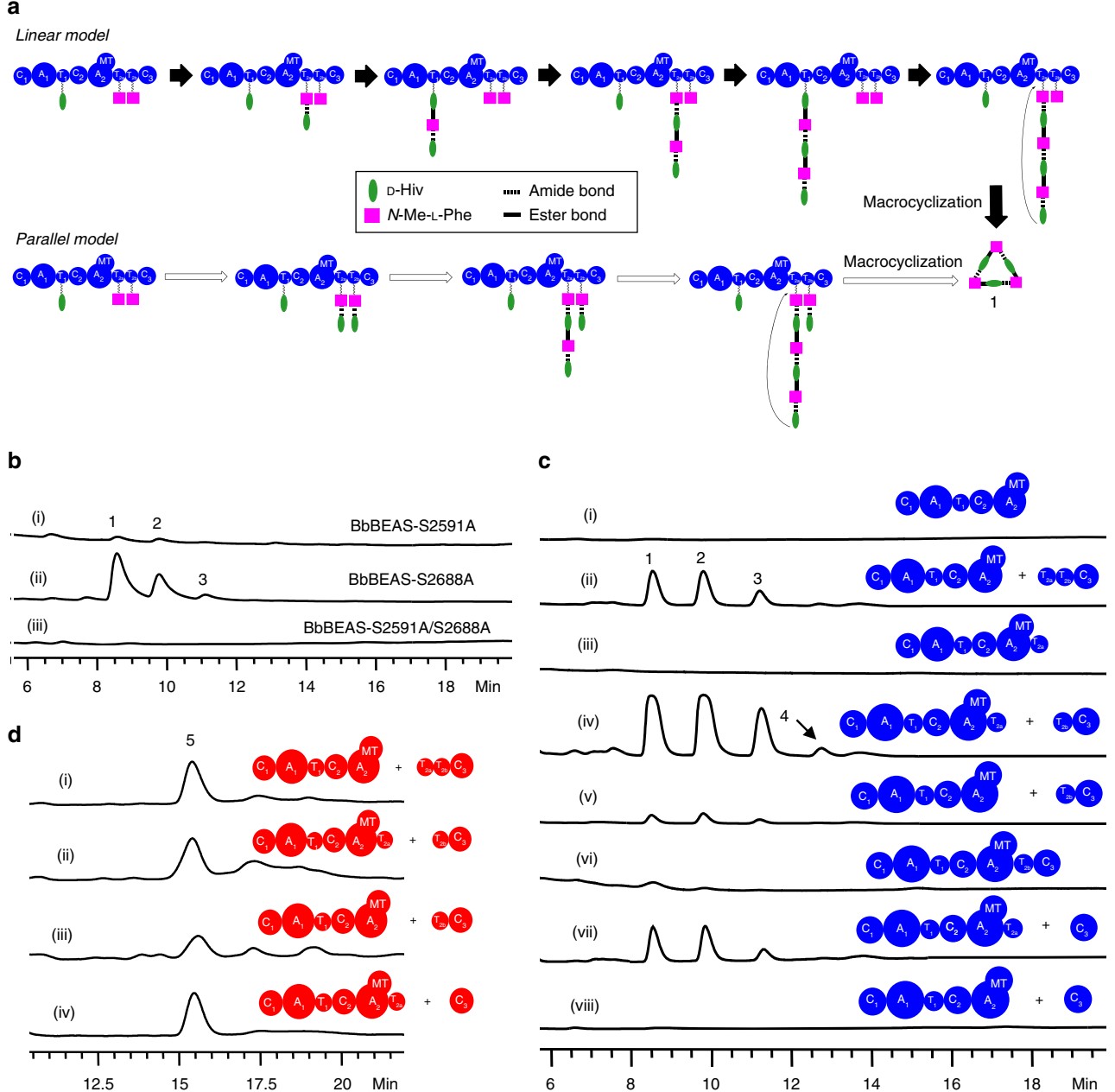

**Figure 2 | Roles of the twin $T_2$ domains in BbBEAS and BbBSLS. (a)** Two possible models for the assembly process of **1**: linear model (top, with filled arrows) and parallel model (bottom, with unfilled arrows). The two precursors and the growing depsipeptide chain are tethered to the T domains through phosphopantetheinyl prosthetic groups. In the linear model, $T_{2a}$ and $T_{2b}$ have the same function of holding the growing depsipeptide chain. **(b)** HPLC analysis of the products of mutant enzymes BbBEAS-S2591A, BbBEAS-S2688A and BbBEAS-S2591A/S2688A in *S. cerevisiae* BJ5464-NpgA. **(c)** HPLC analysis of the effect of removal of $T_{2a}$ and/or $T_{2b}$ on beauvericin biosynthesis. **(d)** HPLC analysis of the effect of removal of $T_{2a}$ and/or $T_{2b}$ on bassianolide biosynthesis.

$C_{1(BbBEAS)}$ lacks the active site H, we mutated D179 to A and expressed this mutant enzyme BbBEAS-D179A in *S. cerevisiae* BJ5464-NpgA. LC–MS analysis revealed the synthesis of beauvericins by BbBEAS-D179A (trace i, Fig. 3e) at $24.2 \pm 1.0 \, mg \, l^{-1}$, indicating that the mutation did not affect the biosynthetic process. Similarly, mutation of H170 and D174 in the conserved motif of $C_1$ of BbBSLS did not interfere with bassianolide biosynthesis (traces ii and iii, Fig. 3e), indicating that mutation of these functionally or structurally important residues does not influence the biosynthesis of **5**. These results clearly indicate that $C_1$ has no condensation activity. To further

probe the role of $C_1$ in beauvericin biosynthesis, we constructed a truncated version of BbBEAS by removing $C_1$. No beauvericins were produced by BbBEAS-$\Delta C_1$. Addition of standalone $C_{1(BbBEAS)}$ into the yeast did not recover beauvericin biosynthesis. SDS–PAGE analysis revealed that BbBEAS-$\Delta C_1$ (308 kDa) was not expressed and the attempt to purify $His_6$-tagged BbBEAS-$\Delta C_1$ using Ni-NTA chromatography from the yeast host failed. Similarly, BbBSLS-$\Delta C_1$ could not be expressed in the yeast either. Thus, removal of $C_1$ resulted in unsuccessful expression of the truncated enzymes, suggesting that the presence of $C_1$ is critical for the expression of BbBEAS and BbBSLS.

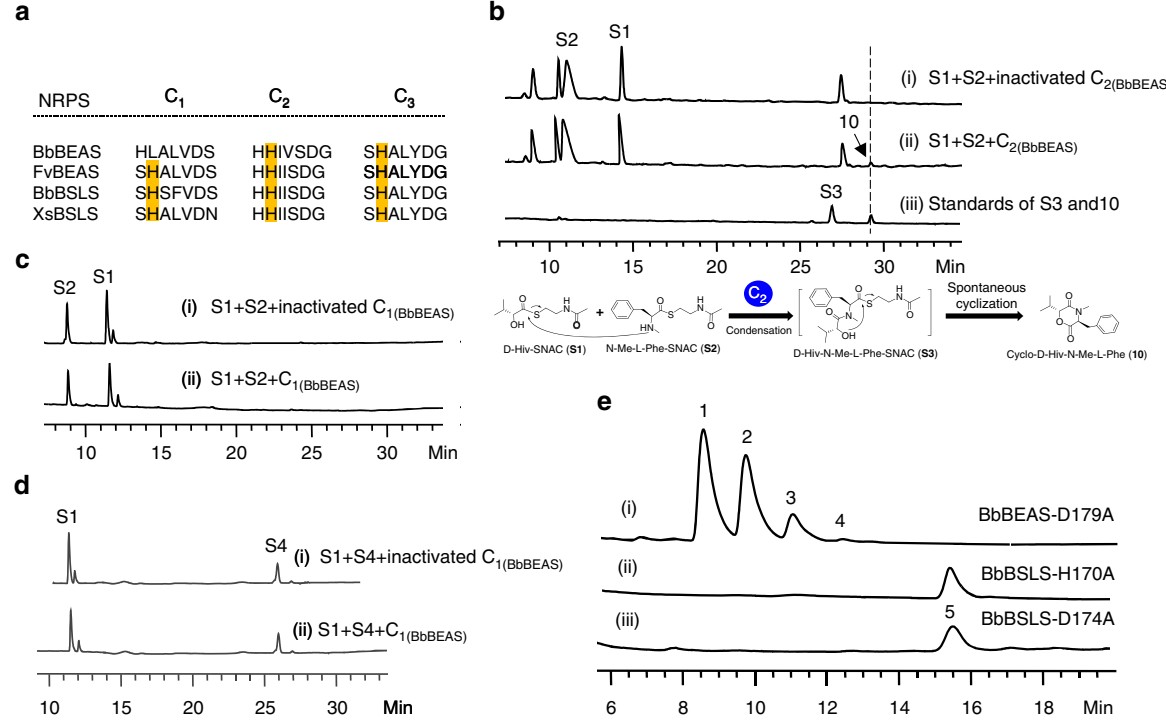

**Figure 3 | Investigation of the condensation activity of $C_1$ and $C_2$.** (**a**) Comparison of the conserved HHxxxDG region in the C domains of four reported fungal iterative NRPSs. The active site H residues are highlighted. (**b**) HPLC analysis of the *in vitro* reaction of $C_{2(BbBEAS)}$ with D-Hiv-SNAC (**S1**) and $N$-Me-L-Phe-SNAC (**S2**). (**c**) HPLC analysis of the *in vitro* reaction of $C_{1(BbBEAS)}$ with **S1** and **S2**. (**d**) HPLC analysis of the *in vitro* reaction of $C_{1(BbBEAS)}$ with **S1** and **S4**. (**e**) HPLC analysis of the products of $C_1$-mutated BbBEAS and BbBSLS in *S. cerevisiae* BJ5464-NpgA.

**The condensation activity of the $C_3$ domain.** To understand how the ester bond is formed, we next investigated the condensation activity of $C_3$. We first removed this domain to yield a truncated variant of BbBEAS. Expression of BbBEAS-$\Delta C_3$ yielded no products (trace i, Fig. 4a). *In trans* addition of standalone $C_{3(BbBEAS)}$ into *S. cerevisiae* BJ5464-NpgA reconstituted the synthesis of beauvericins (26.5 ± 1.3 mg l$^{-1}$, trace ii, Fig. 4a). This suggests that $C_3$ plays an essential role in beauvericin biosynthesis. We next examined the effect of mutation of the conserved H on beauvericin biosynthesis. BbBEAS-H2901A failed to produce beauvericins (traces iii, Fig. 4a), indicating that the condensation activity of $C_3$ is required for the biosynthetic process.

To gain more insight into the role of $C_3$, C-His$_6$-tagged BbBEAS-$\Delta C_3$ was purified from *S. cerevisiae* BJ5464-NpgA and reacted with ATP, SAM, D-Hiv and L-Phe. No beauvericins were detected by LC–MS. Instead, small amounts of **10** and **11** (trace i, Fig. 4b) were produced, which were respectively identified as the cyclic and linear D-Hiv-$N$-Me-L-Phe based on the ESI-MS (Supplementary Fig. 2) and NMR (Supplementary Table 1) as well as a comparison with the chemically prepared standards (traces ii and iii, Fig. 4b). **10** can be hydrolysed in 0.1 N NaOH to yield **11**. Production of **10** and **11** indicated that the biosynthetic process stopped after the formation of the monomer. The same products were observed from the reaction of purified C-His$_6$-tagged BbBEAS-H2901A (trace iv, Fig. 4b), confirming that mutation of H2901 did cause the loss of the condensation ability of $C_3$.

We also overexpressed and purified $C_{3(BbBEAS)}$ from *E. coli* BL21(DE3) (Supplementary Fig. 6). Co-reaction of $C_{3(BbBEAS)}$ and BbBEAS-$\Delta C_3$ with necessary components yielded **1** (trace i, Fig. 4c). (D-Hiv-$N$-Me-L-Phe)$_2$-D-Hiv-SNAC (**S5**) was synthesized from **S1** and **S4** by standalone $C_{3(BbBEAS)}$, then spontaneously hydrolysed to form (D-Hiv-$N$-Me-L-Phe)$_2$-D-Hiv (**14**, Fig. 4d) that

has a molecular mass of 640 (Supplementary Fig. 2). Addition of **S2** into the reaction system did not produce any extra products including **1** (Supplementary Fig. 8). Thus, $C_3$ is responsible for the formation of the ester bond between T$_1$-linked D-Hiv and the monomer D-Hiv-$N$-Me-L-Phe or the dimer (D-Hiv-$N$-Me-L-Phe)$_2$ that is tethered to T$_{2a}$ or T$_{2b}$.

The same experiments were applied to BbBSLS. BbBSLS-$\Delta C_3$ and BbBSLS-H2861A failed to synthesize **5** (traces i and iii, Fig. 4e). *In trans* addition of standalone $C_{3(BbBSLS)}$ recovered bassianolide biosynthesis (traces ii and iv, Fig. 4e). *In vitro* reactions of ATP, SAM, D-Hiv and L-Leu with purified BbBSLS-$\Delta C_3$ (trace i, Fig. 4f) or BbBSLS-H2861A (trace ii, Fig. 4f) did not yield **5**, but the linear (**12**) and cyclic D-Hiv-$N$-Me-L-Leu (**13**) (traces i–iv, Fig. 4f), respectively. Addition of $C_{3(BbBSLS)}$ into the reaction system reconstituted bassianolide biosynthesis (trace ii, Fig. 4c). These results further confirmed that $C_3$ plays an essential role in forming the ester bond during the biosynthesis of **1**–**5**. Removal or inactivation of $C_3$ did not affect the functions of the upstream domains, but stopped the biosynthetic process at the monomer stage.

**Identification and reprogramming of chain length control.** With the understanding of the roles of the twin T$_2$ and three condensation domains, we next attempted to identify the product-length-controlling domain(s) by constructing a series of chimeric enzymes. Since module 1 of BbBEAS and BbBSLS are exchangeable without affecting the product profiles[19], we tested the effects of swapping the C-terminal domains on the product formation. We first constructed an enzyme $C_1$-$A_1$-$T_1$-$C_2$-$A_2$-MT$_{(BbBEAS)}$T$_{2a}$-T$_{2b}$-$C_{3(BbBSLS)}$. Unlike wild-type BbBEAS (trace i, Fig. 5a), $C_1$-$A_1$-$T_1$-$C_2$-$A_2$-MT$_{(BbBEAS)}$T$_{2a}$-T$_{2b}$-$C_{3(BbBSLS)}$ did not synthesize beauvericins, but a new product FX1 (**15**) (trace ii,

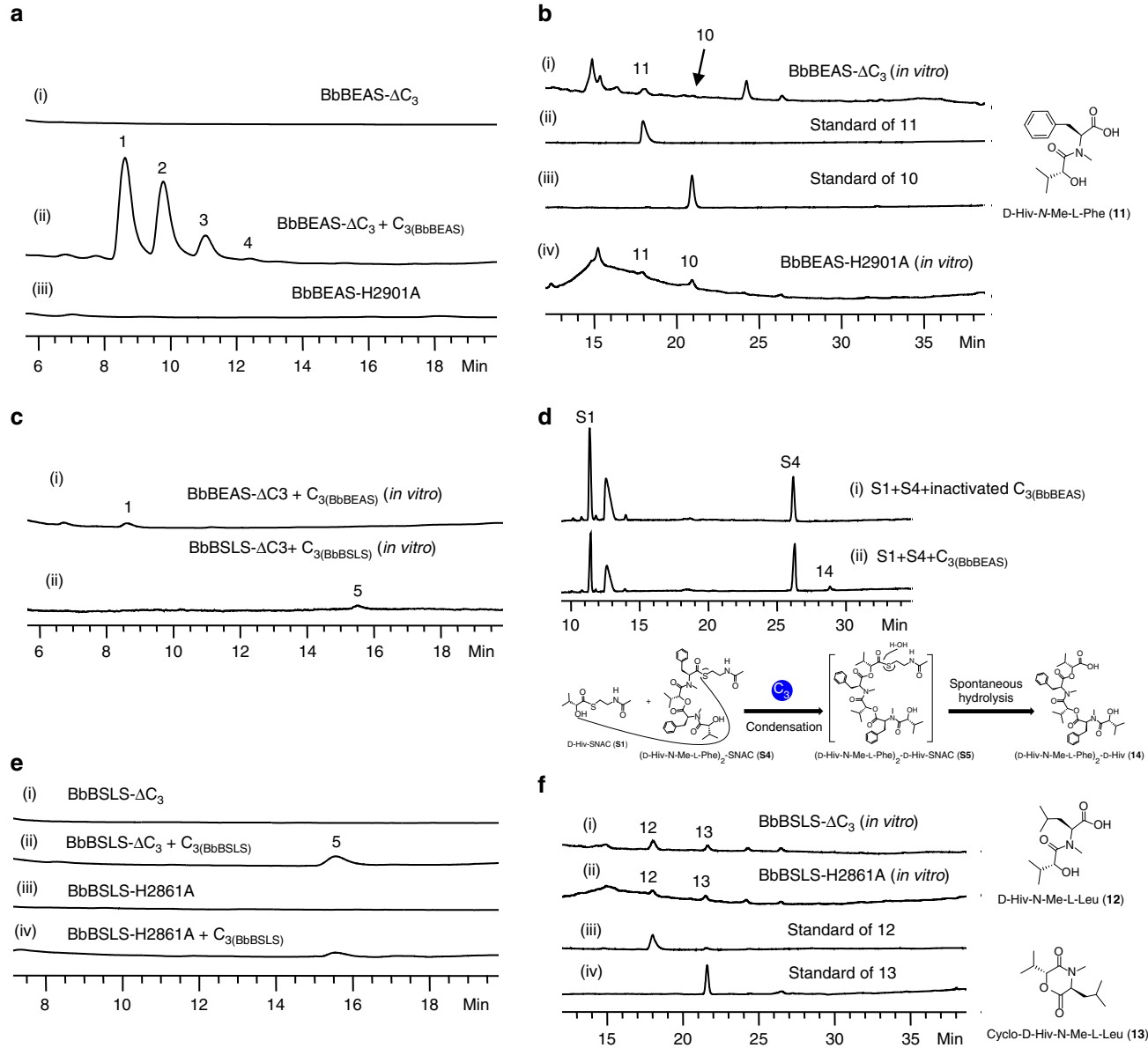

**Figure 4 | Investigation of the ether bond-forming activity of C₃.** (**a**) HPLC analysis of the *in vivo* products of C₃-less or C₃-mutated BbBEAS in *S. cerevisiae* BJ5464-NpgA. (**b**) HPLC analysis of the *in vitro* reactions of C₃-less or C₃-mutated BbBEAS with ATP, SAM, D-Hiv and L-Phe. (**c**) HPLC analysis of the *in vitro* reconstitution of the biosynthesis of **1** and **5** by the co-reactions of C₃-less NRPS and standalone C₃ with ATP, SAM, D-Hiv and L-amino acid (L-Phe for BbBEAS and L-Leu for BbBSLS). (**d**) HPLC analysis of the *in vitro* reaction of C₃(BbBEAS) with D-Hiv-SNAC (**S1**) and (D-Hiv-*N*-Me-L-Phe)₂-SNAC (**S4**). (**e**) HPLC analysis of the *in vivo* products of C₃-less or C₃-mutated BbBSLS in *S. cerevisiae* BJ5464-NpgA. (**f**) HPLC analysis of the *in vitro* reactions of C₃-less or C₃-mutated BbBSLS with ATP, SAM, D-Hiv and L-Leu.

Fig. 5a) with a molecular mass of 1,044 (Supplementary Fig. 2) at $3.2 \pm 0.6$ mg l$^{-1}$. The molecular formula of **15** was determined to be $C_{60}H_{76}N_4O_{12}$ by HR-ESI-MS (Supplementary Fig. 9). The structure was established as a cyclic tetramer (Fig. 5b) based on the one-dimensional (Supplementary Table 1) and two-dimensional NMR data (Supplementary Fig. 10). This was further confirmed by chemical hydrolysis of **15** in 0.1 N NaOH, which yielded the monomer **11** (Supplementary Fig. 11). We then made $C_1$-$A_1$-$T_1$-$C_2$-$A_2$-MT-$T_{2a(BbBEAS)}$-$T_{2b}$-$C_{3(BbBSLS)}$ and $C_1$-$A_1$-$T_1$-$C_2$-$A_2$-MT-$T_{2a}$-$T_{2b(BbBEAS)}$-$C_{3(BbBSLS)}$. Both enzymes produced **15** as the sole product (traces iii and iv, Fig. 5a). These results revealed that replacement of $C_{3(BbBEAS)}$ with $C_{3(BbBSLS)}$ is sufficient to reprogramme chain length control in BbBEAS. The same results were obtained for BbBSLS (traces i–iv,

Fig. 5c). $C_1$-$A_1$-$T_1$-$C_2$-$A_2$-MT-$T_{2a}$-$T_{2b(BbBSLS)}$-$C_{3(BbBSLS)}$ generated **8** (Supplementary Fig. 12) as the only product. Therefore, $C_3$ was unambiguously identified as the chain-length-controlling domain.

C-terminal TE, R, Cy or C is often involved in the cyclization and concomitant release of the peptide chain from the end of the NRPS assembly lines[20]. Since no TE, Cy or R domains are present in BbBEAS and BbBSLS, we considered that $C_1$ and $C_3$ might be involved in the macrocyclization. To probe the cyclization activity and specificity of $C_3$, three depsipeptidyl-SNACs that mimic the linear monomer, dimer and trimer intermediates tethered to $T_{2a}$ or $T_{2b}$ were reacted with $C_{3(BbBEAS)}$. **S3** is unstable and can undergo quick spontaneous cyclization to yield **10** in the reaction buffer (traces i and ii, Fig. 5d). Addition of $C_{3(BbBEAS)}$ into the

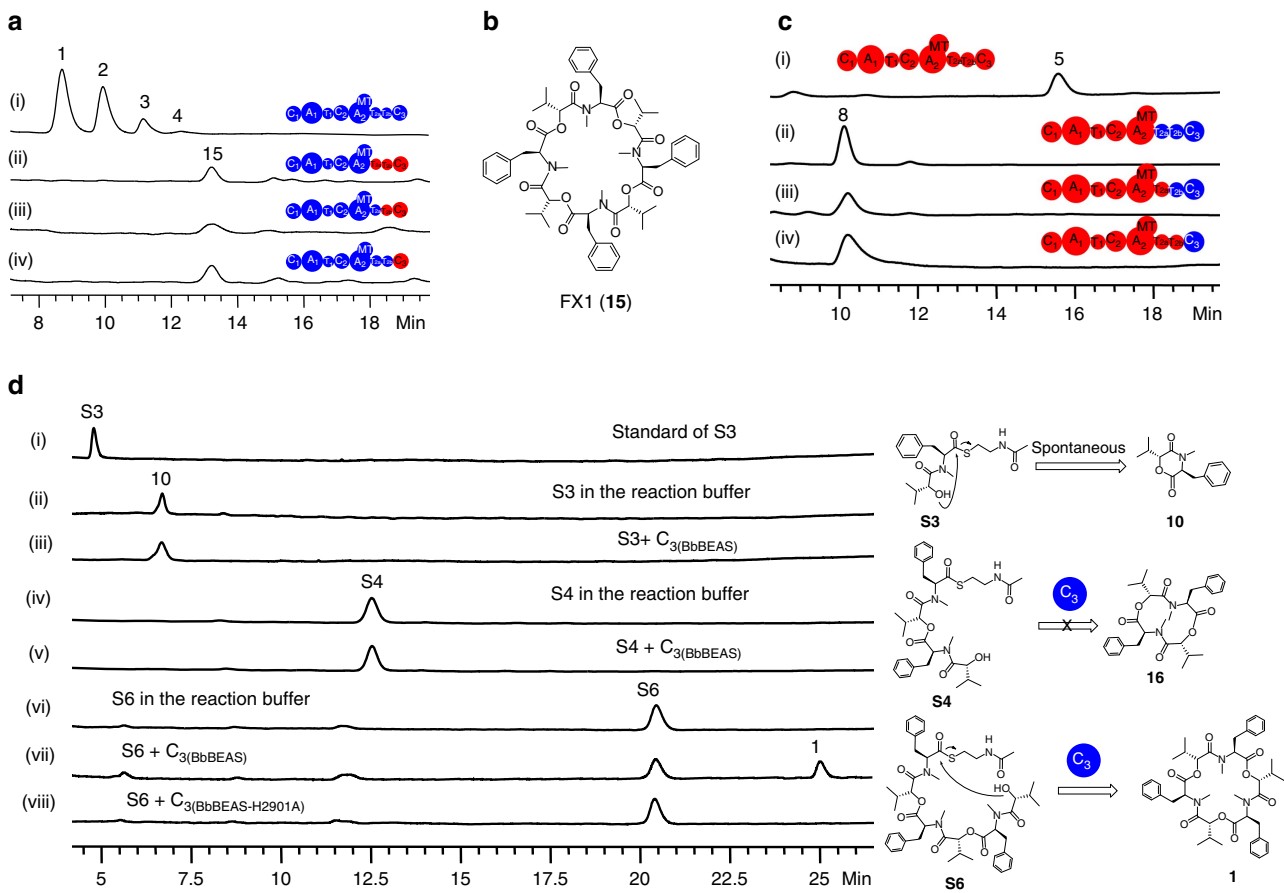

**Figure 5 | Identification and reprogramming of chain length control.** (**a**) HPLC analysis of the products of wild-type BbBEAS and its engineered versions in *S. cerevisiae* BJ5464-NpgA. (**b**) Structure of the new tetrameric product FX1 (**15**) generated by reprogramming BbBEAS. (**c**) HPLC analysis of the products of wild-type BbBSLS and its engineered versions in *S. cerevisiae* BJ5464-NpgA. (**d**) HPLC analysis of the *in vitro* reactions of depsipeptidyl-SNACs with $C_{3(BbBEAS)}$. Three SNAC derivatives including D-Hiv-*N*-Me-L-Phe-SNAC (**S3**), (D-Hiv-*N*-Me-L-Phe)$_2$-SNAC (**S4**) and (D-Hiv-*N*-Me-L-Phe)$_3$-SNAC (**S6**) were used to mimic the intermediates in beauvericin biosynthesis.

system did not cause observable changes in the formation rate of **10** (trace iii, Fig. 5d). The dimer substrate **S4** is stable in the buffer (trace iv, Fig. 5d), but was not cyclized by $C_{3(BbBEAS)}$ to yield **16** (trace v, Fig. 5d). The trimer substrate (D-Hiv-*N*-Me-L-Phe)$_3$-SNAC (**S6**) mimics the full-length intermediate. This substrate is stable in the reaction buffer (trace vi, Fig. 5d). Incubation of **S6** with $C_{3(BbBEAS)}$ yielded **1** (trace vii, Fig. 5d), while $C_{3(BbBEAS-H2901A)}$ failed to cyclize it (trace viii, Fig. 5d). The same reactions were applied to $C_{1(BbBEAS)}$ but no cyclization products **16** and **1** were detected. These *in vitro* reactions further confirmed that $C_3$ catalysed the macrocyclization in beauvericin biosynthesis. Mutation of the active site residue H in $C_{3(BbBEAS)}$ to A abolished its macrocyclization activity, indicating that this domain uses the same active site for the condensation and macrocyclization, both leading to the formation of an ester bond (intermolecular or intramolecular) between the carboxyl of L-Phe and hydroxyl of D-Hiv. Moreover, $C_3$ only cyclizes the mature depsipeptide intermediate. This property is critical for chain length control, which also explains why **10** and **16** are not produced by the beauvericin-producing strain *B. bassiana* and the yeast strain that expresses BbBEAS.

**_In vitro_ total biosynthesis of 1 using the $C_2$ and $C_3$ domains.** Our results revealed that $C_2$ and $C_3$ work collaboratively to generate a linear full-length intermediate that is subsequently

cyclized and released by $C_3$. We next attempted to use these two domains for *in vitro* total biosynthesis of **1** from the beginning precursors. To this end, we reacted $C_{2(BbBEAS)}$ and $C_{3(BbBEAS)}$ with the two mimicking substrates **S1** and **S2**, while the control contains the same components except the two domains were inactivated. The reactions were subjected to LC–MS analysis. As shown in Fig. 6a, there is a small product peak at 25.8 min that had a $[M + H]^+$ peak at $m/z$ 642 (Supplementary Fig. 2). Extraction of the ion chromatogram at $m/z$ 642 (i, Fig. 6a) from the negative control (left panel) and the reaction (right panel) revealed that this product was only formed in the reaction, which has the same molecular mass and retention time (Fig. 4d) as **S4**. The extracted ion chromatogram (EIC) at $m/z$ 784 clearly showed the synthesis of **1** (ii, Fig. 6a). Similarly, EICs at $m/z$ 262, 641 and 903 indicated the formation of **10** (iii), **14** (iv) and **S6** (v) in the reaction mixture. We further enlarged the high-performance liquid chromatography (HPLC) traces of the negative control (i, Fig. 6b), the reaction (ii, Fig. 6b) and the standard of **1** (iii, Fig. 6b), which clearly revealed that **1** was synthesized from the two beginning precursors **S1** and **S2** by the two isolated domains. Furthermore, the peak of **S6** was also observed, which showed the same retention time as the extracted ion peak shown in trace v of Fig. 6a. These results indicated the success of *in vitro* total biosynthesis of **1** by $C_2$ and $C_3$ and showed a series of biosynthetic intermediates or their hydrolysed products.

# a

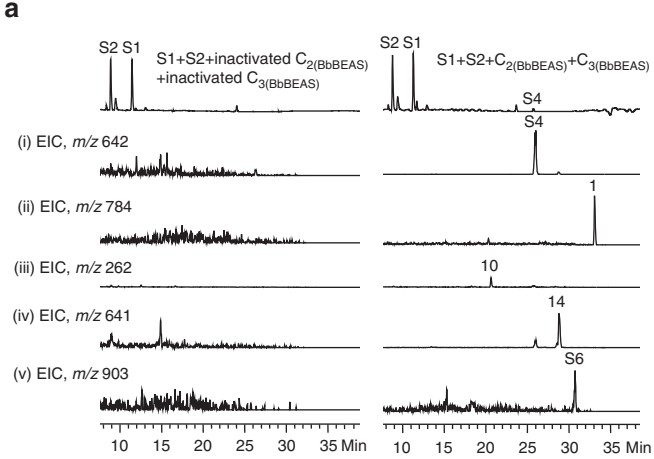

# b

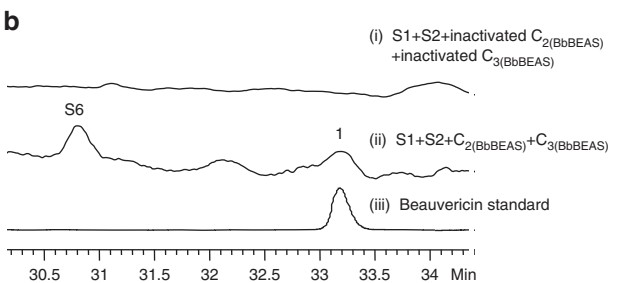

**Figure 6 | *In vitro* total biosynthesis of 1 using C₂ and C₃ domains.**
(**a**) LC–MS analysis of the *in vitro* reaction of $C_{2(BbBEAS)}$ and $C_{3(BbBEAS)}$ with **S1** and **S2**. The left panel is the negative control with inactivated $C_2$ and $C_3$, while the right is the actual reaction. The HPLC traces are shown on the top, followed by the EICs of the intermediates and final product. (i)–(v) are EICs at $m/z$ 642 (i), 784 (ii), 262 (iii), 641 (iv) and 903 (v), respectively.
(**b**) Enlarged view of the region of 30.1–34.4 min of the HPLC traces of the negative control (i), reaction (ii) and the standard of **1** (iii).

## Discussion

Fungal NRPs represent an important group of bioactive natural products. BbBEAS and BbBSLS are two fungal iterative NRPSs that are involved in the biosynthesis of the anticancer NRPs **1–5**. However, the assembly process of these cyclooligomer depsipeptides is unclear, largely due to the poorly understood domains including $T_{2a}$, $T_{2b}$, $C_1$, $C_2$ and $C_3$ in these modular enzymes. We used a combination of enzyme dissection, domain swapping, site-directed mutagenesis and *in vitro* enzymatic reactions to study BbBEAS and BbBSLS. This work elucidates fungal NRPSs' chain elongation and length control strategy. Highly consistent results were obtained from both BbBEAS and BbBSLS, suggesting that this family of fungal iterative NRPSs follows a general biosynthetic mechanism, which is different from the approach used by bacterial iterative NRPSs.

A typical NRPS module contains a C domain, an A domain and a T domain for chain elongation. Co-existence of two T domains in module 2 of fungal iterative NRPSs not only represents an interesting structural feature, but also raises questions about the roles of these T domains in fungal NRP biosynthesis. There are several examples of twin carrier proteins in a megasynthase. For example, it was previously reported that module 6 of the leinamycin polyketide synthase from *Streptomyces atroolivaceus* S-140 contains twin acyl carrier protein (ACP) domains that are separated by a MT domain. Either is sufficient for the biosynthesis, and $ACP_{6-2}$ is preferred[21]. In our work, we found that $T_{2a}$ plays

a major role in the formation of the NRPs. Two possible biosynthetic models have been previously proposed to explain the biosynthetic process of fungal NRPs based on how the depsipeptide chain is elongated and how the twin $T_2$ domains are involved in the assembly process. We specifically mutated the key Ser residue in these T domains that is required for the 4′-phosphopantetheinylation. Inactivation of either $T_2$ domain did not abolish the formation of the corresponding NRPs, but affected the yield to different extents. Similar results were observed when one of the $T_2$ domains was removed, providing solid evidence against the parallel model (Fig. 2a), which requires the functions of both $T_{2a}$ and $T_{2b}$. Thus, it is hypothesized that the biosynthesis of 1–5 proceeds through the linear model (Fig. 2a). This was confirmed by the *in vitro* reactions of $C_{2(BbBEAS)}$ and $C_{3(BbBEAS)}$, which demonstrated that $C_2$ forms the amide bond and $C_3$ catalyses the synthesis of the ester bond. Co-reactions of $C_{2(BbBEAS)}$ and $C_{3(BbBEAS)}$ with D-Hiv-SNAC and N-Me-L-Phe-SNAC reconstituted the *in vitro* biosynthesis of **1** with only two condensation domains. Furthermore, some of the intermediates or hydrolysis products were observed, including cyclo-D-Hiv-N-Me-L-Phe, (D-Hiv-N-Me-L-Phe)₂-SNAC, (D-Hiv-N-Me-L-Phe)₂-D-Hiv (hydrolysis product) and (D-Hiv-N-Me-L-Phe)₃-SNAC, which provides direct evidence for the alternate incorporation approach described in the linear biosynthetic model (Fig. 2a).

$C_3$ is a crucial domain that integrates three functions, including condensation, chain length control and macrocyclization. While it acts like normal condensation domains to form the ester bond in the depsipeptide chain, this domain is actively involved in chain elongation and catalyses the macrocyclization of the mature intermediate for product release. It is different from $C_T$ domains in fungal noniterative NRPSs that only perform cyclization[22] and Cy domains that form oxazoline or thiazoline rings using the D residues in the conserved motif DXXXXD[23,24]. Bacterial iterative NRPSs typically use a C-terminal TE domain to cyclize and release the final products. The enterobactin synthetase is such an enzyme whose TE plays a key role in the oligomerization[25]. The monomer intermediate is transferred from the T domain of EntF (C-A-T-TE) to the active site Ser residue of the TE, which allows the T domain to be used for the formation of the next monomer. The TE catalyses the oligomerization and holds the growing peptide chain until it reaches the full length for macrocyclization. A similar TE was observed in the tyrocidine[12] and gramicidin NRPSs[13]. It was also reported that cyclooligomerization can be catalysed through ATP-dependent condensation reactions by NRPS-independent siderophore synthetases, such as DesD[26], PubC[27] and BibC[28] that participate in the biosynthesis of desferrioxamine, putrebactin and bisucaberin, respectively.

Here we show that fungal iterative NRPSs use a different strategy for chain elongation and product length control. Bacterial NRPSs use an oligomerization approach through their C-terminal TE for product chain elongation. The cyclooligomerization by bacterial iterative NRPSs involves acylation and deacylation of the active site serine residue of the C-terminal TE. The monomer intermediate is formed by upstream domains and transferred from a T domain to the Ser of TE. Additional units of the monomer are then added to the monomer-O-TE intermediate to elongate the peptide chain to the full length for product cyclization and release. Therefore, bacterial iterative NRPSs oligomerize a monomer unit that is synthesized from the precursors for chain elongation, and the growing oligomer intermediate is formed at and always tethered to the TE. In contrast, fungal iterative NRPSs use an alternate incorporation approach to extend the peptide chain. The growing depsipeptide chain in the biosynthesis of **1–5** swings between $T_1$ and the twin $T_2$ domains. The monomer is not used as a unit for chain

elongation at all. $C_3$ needs to work with the other condensation domain ($C_2$) to alternately incorporate the two precursors (D-Hiv and N-Me-L-Phe or N-Me-L-Leu). Therefore, bacterial iterative NRPSs adopt a product assembly process that is more like the parallel model in Fig. 2a, while fungal iterative NRPSs use a linear biosynthetic route.

In the linear model (Fig. 2a), $C_{3(BbBEAS)}$ forms an ester bond between $T_1$-tethered D-Hiv and $T_{2a}$- or $T_{2b}$-tethered D-Hiv-N-Me-L-Phe and (D-Hiv-N-Me-L-Phe)$_2$ in beauvericin biosynthesis, while $C_2$ catalyses the formation of an amide bond between $T_{2a}$- or $T_{2b}$-tethered N-Me-L-Phe and $T_1$-tethered D-Hiv, D-Hiv-N-Me-L-Phe-D-Hiv and (D-Hiv-N-Me-L-Phe)$_2$-D-Hiv. $C_{3(BbBSLS)}$ has the same function except that N-Me-L-Leu is used. Both $C_2$ and $C_3$ work collaboratively to elongate the depsipeptide chain. Once $C_2$ catalyses the last condensation step to yield the full-length depsipeptide chain, $C_3$ acts as a decision maker to determine and inform the biosynthetic machinery of whether further elongation is required. For instance, when a hexadepsipeptide intermediate is synthesized, $C_{3(BbBEAS)}$ will shut down the elongation line and perform the macrocyclization. Instead, $C_{3(BbBSLS)}$ will continue the elongation to get an octadepsipeptide intermediate and then conduct the macrocyclization and product release. When $C_{3(BbBEAS)}$ was substituted for the $C_3$ in BbBSLS, it continued to condense D-Hiv and the hexadepsipeptide intermediate (D-Hiv-N-Me-L-Phe)$_3$, and $C_{2(BbBEAS)}$ will continue to form an additional amide bond to get an octadepsipeptide. Cyclization of this intermediate yielded the new product **15**. Similarly, substitution of $C_{3(BbBSLS)}$ with its counterpart in BbBEAS switched BbBSLS from an octadepsipeptide synthetase to a hexadepsipeptide synthetase and yielded **8**. Thus, $C_2$ is flexible and can catalyse extra or fewer condensations according to the 'command' of $C_3$. While $C_3$ is strict with the number of condensation, it is flexible enough to condense 'unnatural' precursors. Consequently, reprogramming of BbBEAS and BbBSLS can be conveniently and readily achieved by swapping the $C_3$ domain. Our work thus provides an unprecedented tool for engineering fungal iterative NRPSs to yield 'unnatural' cyclooligomer depsipeptides with varied chain lengths. This work also presents in vitro total biosynthesis of a fungal NRP using only two condensation domains.

## Methods

**Analysis and purification of compounds.** Compounds were analysed at 210 nm using an Agilent Eciplse XDB-C18 column (5 μm, 4.6 mm × 250 mm) on an Agilent 1200 HPLC coupled with an Agilent 6130 single quadrupole mass spectrometer. High-resolution mass spectrum of FX1 was collected on an Agilent 6210 LCMS. Four HPLC programmes were used. Programme 1 (for Figs 1c and 3c,d and Figs 4b,d,f and 6): 5–90% acetonitrile–water with 0.1% formic acid from 0 to 30 min, 90–100% acetonitrile–water with 0.1% formic acid from 30 to 35 min and 100% acetonitrile–water with 0.1% formic acid from 35 to 50 min at 1 ml min$^{-1}$. Programme 2 (for Figs 2b–d and 3e and Figs 4a,c,e and 5a,c): 80–100% acetonitrile–water with 0.1% formic acid over 20 min at a flow rate of 1 ml min$^{-1}$. Programme 3 (for Fig. 5d): 50–100% acetonitrile–water with 0.1% formic acid over 30 min at 1 ml min$^{-1}$. Programme 4 (for Fig. 3b): 5–70% acetonitrile–water with 0.1% formic acid over 40 min at 1 ml min$^{-1}$. Compound purification was performed on the same HPLC.

**Strains and plasmids.** E. coli XL1-Blue (Agilent Technologies) was used for routine cloning and pJET1.2 (Fermentas) was used as the cloning vector. E. coli cells were grown in Luria-Bertani (LB) medium at 37 °C. When necessary, 50 μg ml$^{-1}$ ampicillin was added. E. coli BL21(DE3) (Agilent Technologies) cells were used for expression of $C_{1(BbBEAS)}$, $C_{2(BbBEAS)}$, $C_{3(BbBEAS)}$, $C_{3(BbBEAS-H2901A)}$, $C_{3(BbBSLS)}$ and $MT_{(BbBEAS)}$. S. cerevisiae BJ5464-NpgA (MATα ura3-52 his3-Δ200 leu2-Δ1 trp1 pep4::HIS3 prb1 Δ1.6R can1 GAL) was obtained from Dr. Nancy Da Silva at the University of California, Irvine. The strain was maintained on YPD (10 mg l$^{-1}$ yeast extract; 20 mg l$^{-1}$ peptone; 20 mg l$^{-1}$ dextrose) agar plates at 30 °C and used for expression of BbBEAS, BbBSLS, their mutants or truncated variants, and co-expression of the dissected fragments. The E. coli/S. cerevisiae shuttle vectors YEpADH2p-URA3 and YEpADH2p-TRP1 were gifts from Dr. Yi Tang at the University of California, Los Angeles.

**Gene amplification and plasmid construction.** The gene fragments $C_{3(bbBeas)}$, bbBeas-$ΔC_3$, $C_{1(bbBeas)}$, $C_{1(bbBsls)}$, $T_{2a}T_{2b}C_{3(bbBeas)}$, $T_{2b}C_{3(bbBeas)}$, $C_{3(bbBsls)}$, $C_1A_1T_1C_2A_2MT_{(bbBeas)}$, bbBeas-$ΔC_1$, $C_1A_1T_1C_2A_2MTT_{2a(bbBeas)}$, bbBsls-$ΔC_3$, bbBsls-$ΔC_1$, $C_1A_1T_1C_2A_2MT_{(bbBsls)}$, $T_{2a}T_{2b}C_{3(bbBsls)}$, $T_{2b}C_{3(bbBsls)}$ and $C_1A_1T_1C_2A_2MT$-$T_{2a(bbBsls)}$ were amplified by PCR from pDY37 or pDY42 (ref. 14) using Phusion High-Fidelity DNA Polymerase (New England Biolabs) with specific primers (Supplementary Table 2). These PCR products were ligated into the cloning vector pJET1.2 to yield 16 plasmids including pDY83, pDY85, pDY92, pDY93, pDY104, pDY105, pDY106, pDY108, pDY109, pDY111, pDY112, pDY113, pDY135, pDY136, pDY137 and pDY138. These plasmids were confirmed by digestion checks and gene sequencing.

The $C_{3(bbBeas)}$, $T_{2a}T_{2b}C_{3(bbBeas)}$ and $T_{2b}C_{3(bbBeas)}$ inserts were excised from pDY83, pDY104 and pDY105 with NheI and PmlI and ligated into YEpADH2p-URA3 between the same sites to generate pDY87, pDY114 and pDY115, respectively. The bbBeas-$ΔC_1$, bbBsls-$ΔC_3$, bbBsls-$ΔC_1$, bbBsls-$ΔT_{2a}T_{2b}C_3$ and bbBsls-$ΔT_{2b}C_3$ inserts were excised from pDY109, pDY112, pDY113, pDY135 and pDY138 with SpeI and PmlI and ligated into YEpADH2p-URA3 between the same sites to generate pDY117, pDY119, pDY121, pDY150 and pDY141. The bbBeas-$ΔC_3$, bbBsls-$C_3$, bbBeas-$ΔT_{2a}T_{2b}C_3$, bbBeas-$ΔT_{2b}C_3$, $T_{2a}T_{2b}C_{3(bbBsls)}$ and $T_{2b}C_{3(bbBsls)}$ inserts were excised from pDY85, pDY92, pDY93, pDY106, pDY108, pDY111, pDY136 and pDY137 with NdeI and PmeI and ligated into YEpADH2p-TRP1 between the same sites to yield pDY88, pDY100, pDY101, pDY116, pDY122, pDY118, pDY140 and pDY139, respectively.

Site-directed mutagenesis in pDY37 and pDY42 that harbour the original bbBeas and bbBsls genes was carried out to construct the mutant plasmids. The PCR conditions were as follows: initial activation of the Phusion High-Fidelity DNA Polymerase for 5 min at 95 °C, followed by 36 cycles of 40 s denaturation at 95 °C, annealing for 40 s at 63 °C and extension for 15.5 min at 65 °C. PCR products were treated with DpnI (New England Biolabs) for 24 h at 37 °C to remove the templates, ligated and introduced into E. coli XL1-Blue. Colonies were subjected to digestion checks and sequencing to confirm the correct mutation. pDY145 (bbBeas-H2901A), pDY149 (bbBeas-D179A), pDY158 (bbBeas-S2591A) and pDY162 (bbBeas-S2688A) were made from pDY37, while pDY151 (bbBsls-H170A), pDY152 (bbBsls-H2861A) and pDY161 (bbBsls-D174A) were made from pDY42. For double mutations, pDY183 (bbBeas-S2591A) was made from pDY162.

Based on the AscI site in bbBeas, the gene fragments (AscI)bbBeas − $ΔT_{2a}$, (AscI)bbBeas with $T_{2a}T_{2b}C_{3(bbBsls)}$, (AscI)bbBeas with $T_{2b}C_{3(bbBsls)}$ and (AscI)bbBeas with $C_{3(bbBsls)}$ were amplified by splicing by overlap extension PCR using pDY37 and pDY42 as the templates. Similarly, (BsrGI)bbBsls with $T_{2a}T_{2b}C_{3(bbBeas)}$, (BsrGI)bbBsls with $T_{2b}C_{3(bbBeas)}$ and (BsrGI)bbBsls with $C_{3(bbBeas)}$ were amplified. These gene fragments were ligated into the cloning vector pJET1.2 to yield seven plasmids including pDY165, pDY188, pDY189, pDY190, pDY191, pDY192 and pDY222. They were confirmed by digestion checks and gene sequencing. The (AscI)bbBeas-$ΔT_{2a}$, (AscI)bbBeas with $T_{2a}T_{2b}C_{3(bbBsls)}$, (AscI)bbBeas with $T_{2b}C_{3(bbBsls)}$ and (AscI)bbBeas with $C_{3(bbBsls)}$ inserts were excised with AscI and PmlI from the corresponding pJET1.2-derived plasmids and ligated into pDY37 between the same sites to generate pDY173, pDY201, pDY204 and pDY203, respectively. The (BsrGI)bbBsls with $T_{2a}T_{2b}C_{3(bbBeas)}$, (BsrGI)bbBsls with $T_{2b}C_{3(bbBeas)}$ and (BsrGI)bbBsls with $C_{3(bbBeas)}$ inserts were excised with BsrGI and PmlI from the corresponding pJET1.2-derived plasmids and ligated into pDY42 between the same sites to generate pDY215, pDY205 and pDY224, respectively.

The gene fragments $C_{1(bbBeas)}$, $C_{2(bbBeas)}$, $C_{3(bbBeas)}$, $C_{3(bbBsls)}$ and $MT_{(bbBeas)}$ were amplified by PCR from the genomic DNA of B. bassiana ATCC 7,159 with Phusion High-Fidelity DNA Polymerase with specific primers (Supplementary Table 2). These gene fragments were ligated into the cloning vector pJET1.2 to yield five plasmids including pFC1, pFC62, pFC2, pFC9 and pZJ134. The gene fragment $C_{3(bbBeas-H2901A)}$ was amplified from pDY145 and ligated into pJET1.2 to yield pFC44. These plasmids were confirmed by digestion checks and gene sequencing. The $C_{1(bbBeas)}$, $C_{3(bbBsls)}$ and $MT_{(bbBeas)}$ inserts were excised from pFC1, pFC9 and pZJ134 with NdeI and BamHI and ligated into pET28a between the same sites to generate pFC3, pFC11 and pJCZ21, respectively. The $C_{2(bbBeas)}$ insert was excised from pFC62 with NdeI and HindIII and ligated into pET28a between the same sites to generate pFC63. The $C_{3(bbBeas)}$ and $C_{3(BbBEAS-H2901A)}$ inserts were excised from pFC2 and pFC44 with NheI and BamHI and ligated into pET28a between the sites to generate pFC4 and pFC46, respectively.

The plasmids constructed in this work are shown in Supplementary Table 3.

**Analysis of the products of the engineered yeast strains.** The NRPSs were expressed in S. cerevisiae BJ5464-NpgA. The correct transformants were selected by autotrophy of uracil and/or tryptophan. Yeast strains harbouring one plasmid were cultured in 50 ml of SC-Ura (or -Trp) dropout medium (6.7 mg l$^{-1}$ yeast nitrogen base; 20 mg l$^{-1}$ glucose; 0.77 mg l$^{-1}$ -Ura or 0.74 mg l$^{-1}$ -Trp dropout supplement) at 30 °C with shaking at 250 rpm. For co-expression experiments, -Trp/-Ura dropout was used. After the OD$_{600}$ value reached 0.6, an equal volume of YP medium (10 mg l$^{-1}$ yeast extract; 20 mg l$^{-1}$ peptone) was added, and the cultures were maintained under the same conditions for an additional 3 days. The cultures were then extracted three times with 100 ml of ethyl acetate and subjected to analysis on an Agilent 1200 HPLC (at 210 nm) coupled with an Agilent 6130 single quadrupole mass spectrometer. The product titres were

calculated from three independent experiments based on the standard curves of purified compounds.

**Protein expression and purification.** C-terminal His$_6$-tagged BbBEAS, BbBSLS and their mutants and truncated variants were expressed in *S. cerevisiae* BJ5464-NpgA for protein purification. For 1 l of yeast culture, the cells were grown at 25 °C in modified YPD medium (with appropriate dropout supplement) that contains 1% dextrose for 72 h. The cells were collected by centrifugation (4,000 rpm for 20 min), resuspended in 20 ml of lysis buffer (50 mM NaH$_2$PO$_4$, pH 8.0, 0.15 M NaCl, 10 mM imidazole) and lysed with sonication on ice. Cell debris was removed by centrifugation (35,000*g* for 1 h at 4 °C). Two millilitres of Ni-NTA agarose resin was incubated with the supernatant at 4 °C for 7 h. The mixture was loaded into a gravity flow column. Buffer A (50 mM Tris-HCl, pH 7.9, 2 mM EDTA, 2 mM DTT) with increasing concentrations of imidazole was used as washing buffers. Purified proteins were concentrated and exchanged into the reaction buffer (50 mM Tris-HCl, pH 8.0). The yields of these modular enzymes were ~0.8 mg l$^{-1}$. N-terminal His$_6$-tagged C$_{1(BbBEAS)}$ (22.5 mg l$^{-1}$), C$_{2(BbBEAS)}$ (35.1 mg l$^{-1}$), C$_{3(BbBEAS)}$ (29.2 mg l$^{-1}$), C$_{3(BbBEAS-H2901A)}$ (22.1 mg l$^{-1}$), C$_{3(BbBSLS)}$ (23.6 mg l$^{-1}$) and MT$_{(BbBEAS)}$ (23.0 mg l$^{-1}$) were expressed with the induction of 200 μM isopropyl β-D-1-thiogalactopyranoside and purified from *E. coli* BL21(DE3) strains harbouring pFC3, pFC63, pFC4, pFC46, pFC11 and pJCZ21, respectively, using a similar Ni-NTA chromatography method.

**Purification and structural characterization of 8 and 15.** To isolate the two products from reprogrammed BbBEAS and BbBSLS, the corresponding cultures were scaled up to 2 l and grown at 30 °C with shaking at 250 rpm for 72 h. The ethyl acetate extracts were dried under reduced pressure and subjected to MCI column chromatography, successively eluted with a gradient of methanol-water (0:100, 20:80, 50:50 and 100:0, v/v). The fractions containing the target compounds were further separated by HPLC, eluted with a gradient of acetonitrile-water (80–100% over 20 min) at a flow rate of 1 ml min$^{-1}$, to yield **8** (10.1 mg) and **15** (2.5 mg). The purified compounds were identified based on the spectral data.

**Chemical preparation of substrates and intermediate products.** Beauvericins (**1–4**) and bassianolide (**5**) were extracted with ethyl acetate from 5 l of fermentation broths of *S. cerevisiae* BJ5464-NpgA/pDY37 and *S. cerevisiae* BJ5464-NpgA/pDY42, respectively. After evaporation of the solvent, the crude extracts were hydrolysed in 20 ml of acetonitrile-water (1:1) containing 0.1 N NaOH for 48 h at 40 °C with stirring. The resultant hydrolytic products were extracted by ethyl acetate, loaded on a silica gel 60 open column, and eluted with a gradient of acetone-hexanes (50, 70 and 100%, v/v) and methanol. The monomer (**11** or **12**)-containing fractions were purified by HPLC, washed with 40% acetonitrile-water at a flow rate of 1 ml min$^{-1}$ to yield 71.2 mg of **11** and 51.6 mg of **12**.

A mixture of 1-ethyl-3-(3-dimethylaminopropyl) carbodiimide hydrochloride (0.1 mmol) and hydroxybenzotriazole (0.1 mmol) was added into 15 ml of tetrahydrofuran containing 0.1 mmol **11** or **12**. After stirring the resulting mixture for 1 h at 40 °C, K$_2$CO$_3$ (0.05 mmol) was added, and the reaction was stirred for overnight at 40 °C. The reaction was then stopped and concentrated by rotary evaporation. The resultant cyclized products **10** and **13** were purified by HPLC. A gradient of acetonitrile-water (5–60% over 30 min) was programmed at a flow rate of 1 ml min$^{-1}$ and 17.6 mg of **10** and 12.2 mg of **13** were purified from the respective reactions.

To prepare D-Hiv-*N*-Me-L-Phe-SNAC (**S3**), **11** (0.1 mmol) and SNAC (0.12 mmol) were added to 1 ml of dimethylformamide in a 10- ml round flask and cooled down to 0 °C. The solution was then treated with diphenylphosphorylazide (0.15 mmol) and triethylamine (0.2 mmol). The reaction mixture was stirred for 3 h before the reaction was stopped and concentrated by rotary evaporation. The resultant monomer-SNAC was purified by HPLC. A gradient of acetonitrile-water (5–90% over 30 min) was programmed at a flow rate of 1 ml min$^{-1}$ to yield 7.7 mg of **S3**. D-Hiv was purchased from Sigma-Aldrich (St Louis, USA), and (D-Hiv-*N*-Me-L-Phe)$_2$ and (D-Hiv-*N*-Me-L-Phe)$_3$ were purchased from the Chinese Peptide Company (Hangzhou, China), which were used to synthesize D-Hiv-SNAC (**S1**), (D-Hiv-*N*-Me-L-Phe)$_2$-SNAC (**S4**) and (D-Hiv-*N*-Me-L-Phe)$_3$-SNAC (**S6**) using the same method for **S3**.

To prepare *N*-Methyl-L-Phe-SNAC (**S2**), L-Phe-SNAC was synthesized first. *t*-Butyloxy carbonyl (Boc)-L-Phe (1 mmol), *N,N*-dicyclohexylcarbodiimide (1 mmol) and 1-hydroxybenzotriazole (1 mmol) were dissolved in 15 ml of trifluoroacetic acid (TFA), followed by adding *N*-acetylcysteamine (1 mmol). After stirring the resulting solution for 45 min at 25 °C, K$_2$CO$_3$ (0.5 mmol) was added and the reaction mixture was stirred for an additional 3 h at 25 °C. After filtration, the solvent was removed *in vacuo*. The resulting residue was dissolved in ethyl acetate and washed once with an equal volume of 10% aqueous NaHCO$_3$. The organic layer was dried by MgSO$_4$, filtered and concentrated *in vacuo*. The crude product was subjected to silica gel column chromatography, eluted with 4% (v/v) MeOH-CHCl$_3$, to afford Boc-L-Phe-SNACs. Boc group was removed by dissolving Boc-L-Phe-SNAC in 50% TFA/CH$_2$Cl$_2$ and stirring at 25 °C for 1 h. After removing solvent, the residue was taken up in a minimal volume of CH$_2$Cl$_2$ and precipitated with ether. The resulting solid was washed twice with ether and dried to afford L-Phe-SNAC.

A typical methylation reaction of L-Phe-SNAC (100 μl) consisted of 6.4 μM MT$_{(BbBEAS)}$, 0.8 mM L-Phe-SNAC and 2.4 mM SAM in 100 mM Tris-HCl buffer (pH 7.5). The reaction mixtures were incubated at 25 °C for 30 min and then quenched with MeOH (50 μl). The mixtures were briefly vortexed and centrifuged at 15,000 rpm for 5 min to remove the precipitated protein before the samples were injected into LC–MS for analysis. The supernatants were analysed by HPLC under 235 nm, eluted with an increasing gradient of acetonitrile (10–15%) in H$_2$O containing 0.1% TFA with a flow rate of 1 ml min$^{-1}$. Purification of **S2** was carried out using the same HPLC method.

***In vitro* enzymatic studies.** For a typical *in vitro* NRPS activity assay, a 400 μl reaction contained 50 mM Tris-HCl buffer (pH 8.0), 2 μM NRPS (or dissected fragments), 5 mM ATP, 25 mM MgCl$_2$, 7.5 mM L-Leu or L-Phe, 2.25 mM D-Hiv and 3 mM SAM. After 3 h incubation at 25 °C, the reactions were quenched with methanol for LC–MS analysis.

To test the condensation activity of C$_2$, **S1** (0.3 mM) and **S2** (0.3 mM) or **S4** (84 μM) were incubated with 30 μM C$_2$ domain in 1 ml of 50 mM Tris-HCl buffer (pH 7.8) at 25 °C for 12 h.

To test the condensation activity of C$_1$, **S1** (0.3 mM) and **S2** (0.3 mM) or **S4** (84 μM) were incubated with 30 μM C$_1$ domain in 1 ml of 50 mM Tris-HCl buffer (pH 7.8) at 25 °C for 12 h.

To test the condensation activity of C$_3$, **S4** (84 μM), **S1** (0.3 mM) and **S2** (0.3 mM, when needed) were incubated with 30 μM C$_3$ domain in 1 ml of 50 mM Tris-HCl buffer (pH 7.8) at 25 °C for 12 h.

To test the macrocyclization activity of C$_1$ and C$_3$, the SNAC substrates **S3**, **S4** or **S6** (75 μM) were, respectively, incubated with 30 μM C$_1$ or C$_3$ domain or their mutants in 100 μl of 50 mM Tris-HCl buffer (pH 7.8) at 25 °C for 4 h.

For *in vitro* total biosynthesis, **S1** (1 mM) and **S2** (1 mM) were incubated with C$_2$ domain (50 μM) and C$_3$ domain (50 μM) in 2 ml of 50 mM Tris-HCl buffer (pH 7.8) at 25 °C for 12 h.

**Data availability.** The authors declare that the data supporting the findings of this study are available within the article and its Supplementary Information Files. The GenBank accession numbers of BbBEAS and BbBSLS are ACI30655 and ACR78148, respectively. All other data supporting the findings of this study are available from the corresponding author on reasonable request.

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

## Acknowledgements

This work was supported by a Utah State University Research Catalyst grant and grants from the National Natural Science Foundation of China (31170763, 31470787). We are grateful to Dr. Yi Tang, the University of California at Los Angeles, for the YEpADH2p vectors, and Dr. Nancy Da Silva, the University of California at Irvine, for the *S. cerevisiae* BJ5464-NpgA strain used in this research.

## Author contributions

J.Z. conceived of the overall idea and designed the experiments. D.Y. conducted the plasmid construction, protein expression and purification in *S. cerevisiae* and enzymatic studies on the intact and dissected enzymes. F.X. carried out the fermentation of engineered strains, protein expression and purification of $C_1$, $C_3$ and MT in *E. coli*, as well as LC–MS analysis and structural characterization of the products. S.Z. prepared and characterized the monomers from the hydrolysis of beauvericin and bassianolide. D.Y. and F.X. synthesized the SNAC derivatives. All authors analysed and discussed the results. J.Z., D.Y. and F.X. prepared this manuscript.

## Additional information

**Competing interests:** The authors declare no competing financial interests.

