## [Peer Review File · Nature Communications]

Reviewer #1 (Remarks to the Author)

Manuscript NCOMMS-16-20877 'Decoding and reprogramming fungal iterative nonribosomal peptide synthetases' by Yu et al. is of very high quality, provides data and supported interpretation which will significantly advance the field on NRPS and Biosynthetic Gene Cluster (BGC) research. The manuscript provides clear and novel evidence, from detailed analysis of two fungal NRPS of the mechanisms used to control NRP chain elongation, and chain length. Specifically, the manuscript reveals that iterative fungal NRPS likely used a linear biosynthetic route for NRP formation, involving two T domains (T1 and T2) and two C domains (C2 and C3 in Yu et al.). Moreover, in addition to macrocyclization and condensation, C3 also functions to NRP control chain length. I rarely use the work riveting to describe the manuscript review process, but in this case I make an exception. This was an engrossing manuscript to review. The elegant approach and methodology of Yu et al. will both provide insight into iterative NRPS functionality and form the basis of future strategies by others for unravelling the domain functionality of NRPS. In particular the use of free domains co-expressed with mutant NRPS or site directed mutagenized (SDM)-NRPS will form the basis of many future studies. Finally of special interest is that C domains alone were able to catalyze NRP formation in vitro using SNAC-modified amino acids, and Yu et al. provide what appears to be a readily accessible approach to making NRPs of altered lengths. Although the linear model concept for iterative NRPS functionality is clearly explained in the manuscript, and shown in Figure 2, I would help but think a clearer and more explicit illustration of the model would help reader accessibility. A few minor issues might also improve the manuscript accessibility:

1. Lines 86-88, 93-95 and 100-102 sort of restate the problem three times. This could be edited.
2. Line 194: an ester bond.
3. Line 204: Define term 'mutant enzyme' here for clarity.
4. Line 201: change 'supported' to 'clearly indicate'.
5. Line 238 The same products....
6. Line 245: molecular mass, not weight....and elsewhere too.
7. Line 263: change 'of swap of' to 'of swapping the
8. Line 279: Should 'propose' be 'considered'.
9. Line 290: '....confirmed that C3 catalyzed....'"
10. Line 299: Our results.....
11. Line 339: mutated.
12. Line 403: '....can be conveniently and readily achieved by swapping.....'
13. Line 539: open MCI column??
14. Line 591: 400 μ l
15. Line 593: 3 h

Reviewer #2 (Remarks to the Author)

Decoding and reprogramming fungal iterative nonribosomal peptide synthetases
Yu, D.; Xu, F.; Zhang, S.; Zhan, J.

The work presented in this manuscript supports a distinct mechanism for fungal nonribosomal peptide synthetases. The paper discusses a few intriguing enzymes with unique reactivity and the combinatorial analysis shows depth of knowledge towards the proposed biosynthetic pathway. The combination of removal of whole domains and mutagenesis to probe structural significance provides a valid foundation for the linear model mechanism. A few comments and requests for revisions are as follows.

Major Comments:

1. Most of the MS data is not high-resolution and the quality of the manuscript would be significantly increased with high resolution MS data. The only high-resolution MS data presented is

for the new compound FX1. Also, the SI Figure legend implies that all the traces are synthetic, but upon closer reading/observation one can infer that the authors mean biosynthesized. This could be specified a little more clearly.

2. The authors mention the previous characterization of 1 (with reference) but neglect to reference their structural characterization of 5.
3. The identification of the serine residues that act as the phosphopantetheine attachment sites was interesting because the alignment in SI Figure 3 makes it appear that there were numerous other potential serines to test in the near vicinity. It may be beneficial to show other NRPS's that are not as similar, but contain these conserved/aligned serines. Otherwise, the authors should be more descriptive about how these particular serines were chosen.
4. The authors mention the conserved HHxxxDG motif and describe the significance of the second H and D, but it is still unclear to the reader why the first H and G are conserved. If this is still not known, then state that. Otherwise, it may be worth mentioning the significance of at least the first H because C3 has a serine instead.
5. Lines 190-193 discuss the conversion of S1 and S2 to S3, then spontaneous cyclization to 10. Was S3 ever isolated or observed in the HPLC trace? Figure 3b has a peak that may potentially be interpreted as a shifted S3, but this peak is also present in the control (inactivated C2).
6. Lines 201-210 describe the overall characterization of the C1 reaction, but only mutagenesis was performed. A follow-up experiment would be to remove this domain (like what was performed for characterization of all the other domains) and ascertain its significance.
7. Lines 241-245 describe a successive reaction of S1 and S4 to produce S5, which is spontaneously cyclized to form 14. Was S5 isolated or observed in reaction traces?
8. The new product FX1 (15) should have full NMR characterization, and this should include 2D NMR data.

Minor Comments:

9. Line 186, should say, "As the C-terminal condensation domain..."
10. Line 194, "...could not form an ester bond..."
11. Line 238, "The same products were observed..."
12. Line 343, "... the parallel model (Fig. 2a) which requires..."
13. Line 344, "Thus, it is hypothesized that the biosynthesis of 1-5 proceeds through the linear model (Fig. 2a)."
14. Line 396, "When C3(BbBSLS) was substituted for the C3..."
15. Line 445, "E. coli BL21(DE3) (Agilent) were used for expression..."
16. Line 471, "...was carried out to construct the mutant plasmids."
17. Line 517, "...6130 single quadrupole mass spectrometer."
18. Line 556, "...added and the reaction was stirred for overnight at..."
19. Line 572, "hydroxybenzotriazole"

Reviewers' comments:

Reviewer #1 (Remarks to the Author):

Manuscript NCOMMS-16-20877 'Decoding and reprogramming fungal iterative nonribosomal peptide synthetases' by Yu et al. is of very high quality, provides data and supported interpretation which will significantly advance the field on NRPS and Biosynthetic Gene Cluster (BGC) research. The manuscript provides clear and novel evidence, from detailed analysis of two fungal NRPS of the mechanisms used to control NRP chain elongation, and chain length. Specifically, the manuscript reveals that iterative fungal NRPS likely used a linear biosynthetic route for NRP formation, involving two T domains (T1 and T2) and two C domains (C2 and C3 in Yu et al.). Moreover, in addition to macrocyclization and condensation, C3 also functions to NRP control chain length. I rarely use the work riveting to describe the manuscript review process, but in this case I make an exception. This was an engrossing manuscript to review. The elegant approach and methodology of Yu et al. will both provide insight into iterative NRPS functionality and form the basis of future strategies by others for unravelling the domain functionality of NRPS. In particular the use of free domains co-expressed with mutant NRPS or site directed mutagenized (SDM)-NRPS will form the basis of many future studies. Finally of special interest is that C domains alone were able to catalyze NRP formation in vitro using SNAC-modified amino acids, and Yu et al. provide what appears to be a readily accessible approach to making NRPs of altered lengths. Although the linear model concept for iterative NRPS functionality is clearly explained in the manuscript, and shown in Figure 2, I would help but think a clearer and more explicit illustration of the model would help reader accessibility. A few minor issues might also improve the manuscript accessibility:

Response: Thanks for supporting the publication of this paper in *Nature Communications*. We appreciate all the comments. As suggested, we have changed Figure 2a by using symbols to illustrate the two possible biosynthetic models, which we believe can help the readers to easily understand these models. We have also solved all the minor issues pointed out by this reviewer.

1. Lines 86-88, 93-95 and 100-102 sort of restate the problem three times. This could be edited.

Response: As suggested, we have deleted the first two and only retained the final one.

2. Line 194: an ester bond.

Response: As suggested, "a ester bond" has been changed to "an ester bond".

3. Line 204: Define term 'mutant enzyme' here for clarity.

Response: As suggested, "mutant enzyme" has been defined as "mutant enzyme BbBEAS-D179A".

4. Line 201: change 'supported' to 'clearly indicate'.

Response: As suggested, "supported" has been changed to "clearly indicate".

5. Line 238 The same products.....

Response: As suggested, "Same products" has been changed to "The same products".

6. Line 245: molecular mass, not weight....and elsewhere too.

Response: As suggested, all "molecular weight" has been changed to "molecular mass" in the manuscript.

7. Line 263: change 'of swap of' to 'of swapping the

Response: As suggested, "of swap of the" has been changed to "of swapping the".

8. Line 279: Should 'propose' be 'considered'.

Response: As suggested, "propose" has been changed to "considered".

9. Line 290: '....confirmed that C3 catalyzed....'"

Response: As suggested, “confirmed that it is C₃ that catalyzes” has been changed to “confirmed that C₃ catalyzed”.

10. Line 299: Our results.....

Response: As suggested, “The above-presented results” has been changed to “Our results”.

11. Line 339: mutated.

Response: As suggested, “mutate” has been changed to “mutated”.

12. Line 403: ‘...can be conveniently and readily achieved by swapping.....’

Response: As the reviewer suggested, “can be conveniently achieved by only swapping” has been changed to “can be conveniently and readily achieved by swapping”.

13. Line 539: open MCI column??

Response: As suggested, we have deleted “open” before MCI column.

14. Line 591: 400 µl

Response: As suggested, “400-µl” has been changed to “400 µl”.

15. Line 593: 3 h

Response: As suggested, “3-h” has been changed to “3 h”.

Reviewer #2 (Remarks to the Author):

Decoding and reprogramming fungal iterative nonribosomal peptide synthetases

Yu, D.; Xu, F.; Zhang, S.; Zhan, J.

The work presented in this manuscript supports a distinct mechanism for fungal nonribosomal peptide synthetases. The paper discusses a few intriguing enzymes with unique reactivity and the combinatorial analysis shows depth of knowledge towards the proposed biosynthetic pathway. The combination of removal of whole domains and mutagenesis to probe structural significance provides a valid foundation for the linear model mechanism. A few comments and requests for revisions are as follows.

Major Comments:

1. Most of the MS data is not high-resolution and the quality of the manuscript would be significantly increased with high resolution MS data. The only high-resolution MS data presented is for the new compound FX1. Also, the SI Figure legend implies that all the traces are synthetic, but upon closer reading/observation one can infer that the authors mean biosynthesized. This could be specified a little more clearly.

Response: Thanks for the comments. Since FX1 is a new compound, we collected high-resolution MS for this compound. Other compounds were identified through a comparison with authentic samples. As the reviewer suggested, we have recorded the high-resolution MS spectra for compounds **1**, **5**, **8** and **11**. Compounds **1** and **5** were generated from the *in vitro* reactions, **8** were produced by the engineered yeast strain through reprogramming of chain length control of BbBSLS, and **11** was obtained by hydrolyzing FX1 with 0.1 N NaOH. In addition, we have also revised the SI figure legends to clearly specify what traces are for biosynthesized compounds and what for chemically synthesized compounds.

2. The authors mention the previous characterization of **1** (with reference) but neglect to reference their structural characterization of **5**.

Response: Thanks for pointing this out. Structural characterization of both **1** and **5** has been reported in Ref. 14 that was published by our group in *Metabolic Engineering* in 2013. As suggested, we have cited this reference at the end of the sentence “Similarly, BbBSLS was found to synthesize **5** (trace iii, Fig. 1c, and Supplementary Fig. 2) from L-Leu and D-Hiv.”

3. The identification of the serine residues that act as the phosphopantetheine attachment sites was interesting because the alignment in SI Figure 3 makes it appear that there were numerous other potential serines to test in the near vicinity. It may be beneficial to show other NRPS's that are not as similar, but contain these conserved/aligned serines. Otherwise, the authors should be more descriptive about how these particular serines were chosen.

Response: Thanks for the suggestion. It is known that T domains in NRPSs contain a conserved motif of (I/L)GG(D/H)SL, in which S is the phosphopantetheine attachment site. As the reviewer suggested, we have revised SI Figure 3 (now SI Figure 5) with the sequence alignment of the twin T2 domains of BbBEAS and BbBSLS with two well studied NRPS T domains. The conserved motif is highlighted and the key Ser residue is boxed.

4. The authors mention the conserved HHxxxDG motif and describe the significance of the second H and D, but it is still unclear to the reader why the first H and G are conserved. If this is still not known, then state that. Otherwise, it may be worth mentioning the significance of at least the first H because C3 has a serine instead.

Response: The second H and D residues are conserved and their functions were investigated in previous studies, including the references (15-17) cited in this paper. The first H is not highly conserved and some NRPSs don't have an H residue at this position. In fact, Ref. 17 showed that mutation of the first H residue of TycC5-T6 to A did not abolish the condensation activity. As shown in Figure 3a, most C domains from the four NRPSs don't contain the first H residue. The exact function of the conserved G residue remains unclear. As the reviewer suggested, we have now stated this in the revised manuscript.

5. Lines 190-193 discuss the conversion of S1 and S2 to S3, then spontaneous cyclization to 10. Was S3 ever isolated or observed in the HPLC trace? Figure 3b has a peak that may potentially be interpreted as a shifted S3, but this peak is also present in the control (inactivated C2).

Response: We synthesized **S3** as a substrate and standard, and found that this compound is not stable in the reaction buffer and quickly spontaneously cyclized to form **10**. Thus, we did not observe **S3** but the cyclized product. The peak at 27.5 min is not **S3**. It was also present in the control and doesn't have the same molecular mass as **S3**.

6. Lines 201-210 describe the overall characterization of the C1 reaction, but only mutagenesis was performed. A follow-up experiment would be to remove this domain (like what was performed for characterization of all the other domains) and ascertain its significance.

Response: Thanks for the suggestion. In fact, we have done the domain removal experiment but did not include the results in the first version of this manuscript. Removal of C1 of BbBEAS abolished the production of beauvericins and we did not detect the expression of BbBEAS- Δ C1. The same result was observed for BbBSLS- Δ C1. These results suggest that C1 is a critical component of BbBEAS and BbBSLS. As the reviewer suggested, we have now mentioned this work in the revised manuscript.

7. Lines 241-245 describe a successive reaction of S1 and S4 to produce S5, which is spontaneously cyclized to form 14. Was S5 isolated or observed in reaction traces?

Response: We did not detect **S5**, but the hydrolyzed product **14**.

8. The new product FX1 (15) should have full NMR characterization, and this should include 2D NMR data.

Response: As suggested, we have added SI Figure 10 to show the COSY and HMBC correlations of FX1.

Minor Comments:

9. Line 186, should say, "As the C-terminal condensation domain..."

Response: As suggested, “As C-terminal condensation domain” has been changed to “As the C-terminal condensation domain”.

10. Line 194, “...could not form an ester bond...”

Response: As suggested, “a ester bond” has been changed to “an ester bond”.

11. Line 238, “The same products were observed...”

Response: As suggested, “Same products” has been changed to “The same products”.

12. Line 343, “... the parallel model (Fig. 2a) which requires...”

Response: As suggested, “the parallel model (Fig. 2a) that requires” has been changed to “the parallel model (Fig. 2a) which requires”.

13. Line 344, “Thus, it is hypothesized that the biosynthesis of 1-5 proceeds through the linear model (Fig. 2a).”

Response: As suggested, “Thus, the linear model (Fig. 2a) is the right one for the biosynthesis of 1-5” has been changed to “Thus, it is hypothesized that the biosynthesis of 1-5 proceeds through the linear model (Fig. 2a)”.

14. Line 396, “When C3(BbBSLS) was substituted for the C3...”

Response: As suggested, “When C₃(BbBSLS) replaced the C₃ in BbBEAS” has been changed to “When C₃(BbBEAS) was substituted for the C3 in BbBSLS”.

15. Line 445, “E. coli BL21(DE3) (Agilent) were used for expression...”

Response: As suggested, “E. coli BL21(DE3) (Agilent) was used for expression” has been changed to “E. coli BL21(DE3) (Agilent Technologies) cells were used for expression.”

16. Line 471, “...was carried out to construct the mutant plasmids.”

Response: As suggested, “was carried out to construct the mutation plasmids” has been changed to “was carried out to construct the mutant plasmids”.

17. Line 517, “...6130 single quadrupole mass spectrometer.”

Response: As suggested, “6130 single quadrupole mass spectrometry” has been changed to “6130 single quadrupole mass spectrometer” in the manuscript.

18. Line 556, “...added and the reaction was stirred for overnight at...”

Response: We have changed “...added and the reaction was stirred for overnight at...” to “...added, and the reaction was stirred for overnight at...”

19. Line 572, “hydroxybenzotriazole”

Response: As suggested, “hedoxybenzotriazole” has been changed to “hydroxybenzotriazole”.

Reviewer #1 (Remarks to the Author)

All suggestions made by me during the original review have been satisfactorily addressed.

Response to Reviewer Comments

REVIEWERS' COMMENTS:

Reviewer #1 (Remarks to the Author):

All suggestions made by me during the original review have been satisfactorily addressed.

Response: Thanks and we are glad that we were able to satisfactorily address all the comments.